



**Combined effects of elevated $p$CO$_2$ and temperature on biomass and carbon**
**fixation of phytoplankton assemblages in the northern South China Sea**
Guang Gao[1,2], Peng Jin[1], Nana Liu[1], Futian Li[1], Shanying Tong[1], David A. Hutchins[3],
and Kunshan Gao[1*]
[1] *State Key Laboratory of Marine Environmental Science, Xiamen University, Xiamen*
*361005, China*
[2] *Marine Resources Development Institute of Jiangsu, Huaihai Institute of Technology,*
*Lianyungang 222005, China*
[3] *Marine Environmental Biology, Department of Biological Sciences, University of*
*Southern California, 3616 Trousdale Parkway, Los Angeles, California 90089, USA*

[*] Correspondence: KunshanGao, Fax: 86-592-2187963, e-mail: ksgao@xmu.edu.cn.



**Abstract**

The individual influences of ocean warming and acidification on marine

organisms have been investigated intensively, but studies regarding the combined
effects of both global change variables on natural marine phytoplankton assemblages
are still scarce. Even fewer studies have addressed possible differences in the
responses of phytoplankton communities in pelagic and coastal zones to ocean
warming and acidification. We conducted shipboard microcosm experiments at both
off-shore (SEATS) and near-shore (D001) stations in the northern South China Sea
(NSCS) under three treatments, low temperature (30.5 $^{o}$C at SEATS and 28.5 $^{o}$C at
D001) and low $p$CO$_2$ (390.0 μatm at SEATS and 420.0 μatm at D001) (LTLC), high
temperature (33.5 $^{o}$C at SEATS and 31.5 $^{o}$C at D001) and low $p$CO$_2$ (390 μatm at
SEATS and 420 μatm at D001) (HTLC), and high temperature (33.5 $^{o}$C at SEATS
and 31.5 $^{o}$C at D001) and high $p$CO$_2$ (1000 μatm at SEATS and 1030 μatm at D001)
(HTHC). Biomass of phytoplankton at both stations were enhanced by HT. HTHC did
not affect phytoplankton biomass at station D001 but decreased it at station SEATS.
At this offshore station HT alone increased daily primary productivity (DPP, μg C (μg
chl $a$)$^{-1}$ d$^{-1}$) by ~64%, and by ~117% when higher $p$CO$_2$ was added. In contrast, HT
alone did not affect DPP and HTHC reduced it by ~15% at station D001. HT
enhanced the dark respiration rate (μg C (μg chl $a$)$^{-1}$ d$^{-1}$) by 64% at station SEATS, but
had no significant effect at station D001, and did not change the ratio of respiration to
photosynthesis at either station. HTHC did not affect dark respiration rate (μg C (μg
chl $a$)$^{-1}$ d$^{-1}$) at either station compared to LTLC. HTHC reduced the respiration to
photosynthesis ratio by ~41% at station SEATS but increased it ~42% at station D001.
Overall, our findings indicate that responses of coastal and offshore phytoplankton
assemblages in NSCS to ocean warming and acidification are contrasting, with the
pelagic phytoplankton communities being more sensitive to these two global change
factors.

**Key words:** ocean acidification, ocean warming, photosynthesis, primary productivity,
respiration, South China Sea

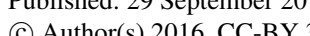




## 1 Introduction

From 1979 to 2012, the mean global sea surface temperature increased at a rate of ~0.12 ℃ per decade, based on in situ data records (IPCC, 2013). Particularly, the warming rate in the South China Sea (~0.26 per decade) from 1982 to 2004 is approximately 2 times faster than the global ocean mean rate (~0.14 per decade) from 1960 to 1990 (Casey and Cornillon, 2001; Fang et al., 2006). It is extremely likely that more than half of the observed increase in global average surface temperature from 1951 to 2010 was caused by the anthropogenic increase in greenhouse gas concentrations and other anthropogenic forcings. Ocean warming is projected to rise approximately by 0.6 ℃ (RCP 2.6) to 2.0 ℃ (RCP 8.5), in the upper 100 m by the end of the 21$^{st}$ century (IPCC, 2013).

Ocean warming is known to affect primary productivity both directly and indirectly. In situ surface chl $a$ declined exponentially with rise of SST (13–23 ℃) in the northeast Atlantic Ocean from latitudes 29 to 63 °N in spring and summer, which was attributed to enhanced stratification and consequent reduced upward transport of nutrients into the upper mixed layer (Poll et al., 2013). The seawater volume-specific primary produtivity also decreased with temperature rise due to lower phytoplanton biomass (Poll et al., 2013). Mesocosm experiments have also demonstrated that ocean warming (an increase of 6 ℃) can decrease phytoplankton biomass in the Baltic Sea (Lewandowska et al., 2012; Sommer et al., 2015). On the other hand, ocean warming did not affect volume-specific primary produtivity in the Baltic Sea (Lewandowska et al., 2012).

Increased atmospheric $CO_2$ is responsible for global warming and the ocean. As a main sink for $CO_2$, the ocean has absorbed approximately 30% of the emitted anthropogenic carbon dioxide (IPCC, 2013). Such a dissolution of $CO_2$ increases seawater $CO_2$ partial pressure and bicarbonate ion levels and decreases pH and carbonate ion concentrations, leading to ocean acidification (OA). By 2100, the projected decline in global-mean surface pH is projected to be approximately 0.065 to



0.31, depending on the RCP scenario (IPCC, 2013). In terms of the South China Sea,
an accelerated trend of ocean acidification has been reported and the rate of pH
decline almost tripled between 1951 and 2000, compared to that between 1840 and
1950 (Liu et al., 2014).
$CO_2$ may be a potentially limiting factor for marine primary productivity
because of the low $CO_2$ level in seawater and the low affinity of the enzyme Rubisco
for dissolved $CO_2$ (Falkowski and Raven, 2013). In addition, $CO_2$ in seawater diffuses
approximately 10,000 times slower than in air, leading to its supply rate being much
lower than the demand of photosynthetic carbon fixation (Raven, 1993; Riebesell et
al., 1993). Although phytoplankton have evolved carbon-concentrating mechanisms
(CCMs) to cope with these problems (Giordano et al., 2005; Raven et al., 2012;
Reinfelder, 2011), increased $CO_2$ concentration may still be beneficial since energy
saved due to down-regulation of CCMs under elevated $CO_2$ can be utilized in other
metabolic processes (Gao et al., 2012a). Early laboratory and shipboard experiments
suggested that increased $CO_2$ indeed could enhance phytoplankton growth rates and
thus marine primary productivity (Riebesell et al., 1993; Hein and Sand-Jensen, 1997;
Schippers et al., 2004). Since then, neutral effects of increased $CO_2$ on growth of
phytoplankton assemblages have also been reported (Tortell and Morel, 2002; Tortell
et al., 2000). Furthermore, ocean acidification can even reduce primary productivity
of surface phytoplankton assemblages when exposed to incident solar radiation (Gao
et al., 2012b). Therefore, the effects of ocean acidification on marine primary
productivity remain controversial and its interactions with other environmental factors,
such as warming, solar UV radiation, hypoxia, etc. are incompletely understood (Gao
et al., 2012a; Häder and Gao, 2015; Mostofa, 2016).
Ocean warming and acidification, both caused by increasing atmospheric $CO_2$,
are proceeding simultaneously. The interactive or combined effects of warming and
OA could be completely different from that of either one stressor (Hare et al. 2007,
Feng et al., 2009; Gao et al., 2012a, Tatters et al. 2013). Several oceanographic cruises
and ship board experiments in the European sector of the Arctic Ocean, showed that



gross primary production increased with $p$CO$_2$ (145–2099 µatm) and the greatest
increase was observed in lower temperature regions, indicating CO$_2$-enhanced
primary production in the European Arctic Ocean is temperature-dependent (Holding
et al., 2015).

The South China Sea (SCS) is located between the equator and 23.8 °N, from

99.1 to 121.1 °E, and is one of the largest marginal seas in the world, with a total area
of about $3.5 \times 10^6$ km$^2$. Therefore, understanding the effects of ocean warming and
acidification on primary production in SCS would help us to define the role of
marginal seas in the global carbon cycle. However, only a very few studies on the
effects of ocean acidification or warming on primary productivity in the SCS have
been reported. Wu and Gao (2010) reported that CO$_2$ enrichment (700 µatm) did not
affect the assimilation number of phytoplankton at a near-shore site in SCS, compared
to the ambient CO$_2$ level (380 µatm). Gao et al. (2012b) demonstrated that increased
$p$CO$_2$ (800 or 1000 µatm) reduced primary productivity in off-shore stations of the
SCS. Therefore, we hypothesized that the effects of ocean acidification or/and
warming on primary productivity in SCS would be site-dependent. None of the
previous studies have examined co-effects of warming and increased CO$_2$ on primary
production in the SCS. In this study, to test this hypothesis we conducted shipboard
microcosm experiments at both near-shore and off-shore stations to determine the
combined effects of ocean warming and acidification on biomass, photosynthetic
carbon fixation, and dark respiration of phytoplankton assemblages in the SCS.
**2 Methods**
2.1 **Experimental setup**
The experiments were conducted at one off-shore station SEATS (17.9963° N,
115.9621° E) and one near-shore station D001 (18.9740° N, 110.7166° E) in the NSCS
(Fig. 1). Surface seawater (0–2 m) was collected before sunrise with a 10 L
acid-cleaned plastic bucket, filtered (180 µm) to remove large grazers and dispensed
into nine microcosms. Microcosms consisted of cylindrical polymethyl methacrylate
tanks (32 L, 0.34 m water depth) with water-jacketed space for circulating cooled



water, allowing 91% photosynthesis active radiation (PAR, 400–700 nm), 63%
ultraviolet-A (UVA, 315–400 nm) and 6% ultraviolet-B (UVB, 280–315 nm)
transmission under incident solar radiation. Two levels of temperature (in situ, in situ
+ 3 $^{o}$C) and $pCO_2$ (ambient, ambient + 610 µatm) were used. There were three
triplicated treatments: low temperature and low $pCO_2$, LTLC; high temperature and
low $pCO_2$, HTLC; high temperature and high $pCO_2$, HTHC. The treatment of low
temperature and high $pCO_2$ was missing due to the lack of microcosms. Microcosm
temperature was controlled with circulating coolers (CTP-3000, EYELA, Japan) and
stable $CO_2$ equilibrium with the sea water (variation of $pCO_2 < 5\%$) was achieved
within 24 hours using a $CO_2$ enricher (CE-100, Wuhan Ruihua Instrument &
Equipment Ltd, China). The incubations were conducted for seven days for station
SEATS (Aug 3$^{rd}$–9$^{th}$ 2012) and six days for station D001 (Aug 14$^{th}$–19$^{th}$ 2012).

**2.2 Solar radiance**

The incident solar radiation was continuously monitored using an Eldonet
broadband filter radiometer (Eldonet XP, Real Time Computer, Germany) that was
fixed at the top of the ship. It measured every second and recorded the means over
each minute.

**2.3 Carbonate chemistry parameters**

The seawater pH in the microcosm was recorded with a pH meter (FE20, Mettler
Toledo, Greifensee, Switzerland) every hour during the first day of incubation and
daily afterwards. The $pCO_2$ in seawater was maintained with the $CO_2$ enricher and
measured by an automated flowing $pCO_2$ measuring system (Model 8050, GO, USA).
Other carbonate system parameters were derived via CO2SYS (Pierrot et al., 2006),
using the equilibrium constants of $K_1$ and $K_2$ for carbonic acid dissociation (Roy et al.,

1993).

**2.4 Chlorophyll *a* analysis**

For the measurement of chlorophyll *a* (chl *a*), 500 mL of seawater were filtered
onto a Whatman GF/F glass fiber filter (25 mm).Then the filter was placed in 5 ml 93%
acetone at -20 ℃ for 24 h. Chl *a* concentration was determined with a fluorometer
(Trilogy, Turner Designs, USA), following the protocol of Welschmeyer (1994).



### 2.5 Primary productivity and dark respiration

Seawater samples taken from each microcosm were dispensed into 50 mL quartz tubes inoculated with 5 μCi (0.185 MBq) $NaH^{14}CO_3$ (ICN Radiochemicals, USA) and incubated for 12 h and 24 h under natural light and day-night conditions. The incubation temperature of every treatment was the same as the corresponding microcosm treatment. After the incubation, the cells were filtered onto a Whatman GF∕F glass fiber filter (25 mm), which was immediately frozen and stored at -20$^o$C for later analysis. In the laboratory, each frozen filter was placed into a 20 mL scintillation vial, exposed to HCl fumes overnight, and dried (55 $^o$C, 6 h) to expel non-fixed $^{14}$C (Gao et al., 2007). Then 3 mL scintillation cocktail (Perkin Elmer®) was added to each vial and incorporated radioactivity was counted by a liquid scintillation counting (LS 6500, Beckman Coulter, USA). The daytime primary productivity (DPP) was defined as the amount of carbon fixation during 12 h incubation. The dark respiration was defined as the difference in amount of carbon fixation between 12 h and 24 h. Carbon fixation over 24 h was taken as daily net primary productivity (NPP). Ratio of respiration to photosynthesis (R/P) was expressed as that of respiratory carbon loss to daytime carbon fixation.

### 2.6 Statistical analyses

Results were expressed as means of replicates ± standard deviation. Data were analyzed using the software SPSS v.21. The data from each treatment conformed to a normal distribution (Shapiro-Wilk, $P > 0.05$) and the variances could be considered equal (Levene's test, $P > 0.05$). One-way ANOVAs were conducted to assess the significant differences in carbonate chemistry parameters, chl $a$, DPP, NPP, dark respiration, ratio of dark respiration to photosynthesis between three treatments. Tukey HSD was conducted for post hoc investigation. Independent samples t-tests were conducted to compare in situ chl $a$, DPP, NPP, dark respiration, and ratio of dark respiration to photosynthesis between both stations. The threshold value for determining statistical significance was $P < 0.05$.

### 3 Results

The incident solar radiation during the experiment was recorded (Table 1). The




daytime (12 h) mean solar radiation ranged from 927 to 1592 μmol photons m$^{-2}$ s$^{-1}$ at
station SEATS while the lowest solar radiance was only 111 μmol photons m$^{-2}$ s$^{-1}$,
with the highest of 1583 μmol photons m$^{-2}$ s$^{-1}$, at station D001. The average of
daytime mean solar radiation over the microcosm incubation at station SEATS was
15.52% higher than that at station D001.
The changes of the seawater carbonate system under different conditions are
shown in Table 2. At station SEATS, an increase of 3 $^{o}$C in temperature (HTLC) did
not alter carbonate parameters except leading to enhanced $CO_3^{2-}$ (Tukey HSD, $P$ =
0.009). HTHC resulted in a significant decrease in $CO_3^{2-}$ (Tukey HSD, $P < 0.001$) and
TA (Tukey HSD, $P = 0.016$) but an increase in $CO_2$ (Tukey HSD, $P < 0.001$)
compared with LTLC. The effect of temperature on carbonate parameters at station
D001 were similar to station SEATS, while HTHC increased $HCO_3^{-}$ (Tukey HSD, $P$ =
0.046) and did not affect TA (Tukey HSD, $P = 0.203$).
The in situ chl $a$ level at station D001 (0.37 $\pm$ 0.05 μg L$^{-1}$) was higher than that at
station SEATS (0.15 $\pm$ 0.02 μg L$^{-1}$) (Independent samples t-test, t = $-$7.483, df = 4, $P$
= 0.002; Fig. 2a). After seven days incubation in the microcosms at station SEATS,
Tukey comparison $(P = 0.05)$ showed that higher temperature (0.46 $\pm$ 0.04 μg L$^{-1}$)
increased chl $a$ compared with LTLC (0.23 $\pm$ 0.03 μg L$^{-1}$) while HTHC (0.07 $\pm$ 0.01
μg L$^{-1}$) reduced it (Fig. 2b). The higher temperature (0.72 $\pm$ 0.07 μg L$^{-1}$) also
increased chl $a$ compared with LTLC (0.41 $\pm$ 0.05 μg L$^{-1}$) at station D001 (Tukey
HSD, $P = 0.001$; Fig. 1b), but with no effect of HTLC (0.41 $\pm$ 0.04 μg L$^{-1}$) (Tukey
HSD, $P = 0.988$).
The in situ DPP at station D001 (49.4 $\pm$ 4.5 μg C L$^{-1}$ d$^{-1}$ or 133.4 $\pm$ 12.1 μg C (μg
chl $a$)$^{-1}$ d$^{-1}$) was dramatically higher than that at station SEATS (5.1 $\pm$ 0.5 μg C L$^{-1}$ d$^{-1}$
or 34.1 $\pm$ 3.1 μg C (μg chl $a$)$^{-1}$ d$^{-1}$), whether normalized to volume of seawater
(Independent samples t-test, t = $-$17.056, df = 4, $P < 0.001$; Fig. 3a) or chl $a$
(Independent samples t-test, t = $-$13.786, df = 4, $P < 0.001$; Fig. 3b). After seven days
incubation in microcosms, the DPP normalized to volume of seawater under HTLC
(33.2 $\pm$ 4.8 μg C L$^{-1}$ d$^{-1}$) at station SEATS was significantly higher than that under
LTLC (9.9 $\pm$ 1.2 μg C L$^{-1}$ d$^{-1}$) and HTHC (6.6 $\pm$ 0.6 μg C L$^{-1}$ d$^{-1}$) (Tukey HSD, $P <$



0.001) while the difference between LTLC and HTHC was insignificant (Tukey HSD,
$P$ = 0.380; Fig. 3c). The pattern at station D001 was similar to SEATS (Fig. 2c). When
DPP was normalized to chl $a$, the higher temperature increased primary productivity
from 43.2 $\pm$ 5.1 to 70.7 $\pm$ 10.1 µg C (µg chl $a$)$^{-1}$ d$^{-1}$ (Tukey HSD, $P$ = 0.014) and
further to 93.9 $\pm$ 8.1 µg C (µg chl $a$)$^{-1}$ d$^{-1}$ (Tukey HSD, $P$ < 0.001) when higher $CO_2$
was combined at station SEATS (Fig. 2d). In contrast, temperature did not affect DPP
(Tukey HSD, $P$ = 0.0924) and HTHC reduced it from 150.3 $\pm$ 4.9 to 128.0 $\pm$ 11.5 µg
C (µg chl $a$)$^{-1}$ d$^{-1}$ (Tukey HSD, $P$ = 0.039) at station D001 (Fig. 3d).

The in situ NPP at stations SEATS and D001 were 3.5 $\pm$ 0.1 µg C L$^{-1}$ d$^{-1}$ (23.2 $\pm$

1.0 µg C (µg chl $a$)$^{-1}$ d$^{-1}$) and 37.4 $\pm$ 3.1 µg C L$^{-1}$ d$^{-1}$ (91.2 $\pm$ 7.5 µg C (µg chl $a$)$^{-1}$ d$^{-1}$)
respectively, which indicates that station D001 has higher NPP, irrespective of
normalizing to volume of seawater (Independent samples t-test, t = −18.998, df = 4, $P$
< 0.001; Fig. 4a) or chl $a$ (Independent samples t-test, t = −15.511, df = 4, $P$ < 0.001;
Fig. 4b). After a seven-day incubation in the microcosms, the higher temperature
increased NPP to 23.9 $\pm$ 5.3 µg C L$^{-1}$ d$^{-1}$ (Tukey HSD, $P$ = 0.001) while HTHC (5.5 $\pm$
0.4 µg C L$^{-1}$ d$^{-1}$) did not change it (Tukey HSD, $P$ = 0.793) compared with LTLC (7.2
$\pm$ 0.8 µg C L$^{-1}$ d$^{-1}$). The effects of temperature and $CO_2$ on NPP at station D001 were
similar to that at station SEATS. When NPP was normalized to chl $a$, the higher
temperature increased NPP from 31.1 $\pm$ 3.5 to 50.9 $\pm$ 11.3 (Tukey HSD, $P$ = 0.044)
and further to 78.3 $\pm$ 5.9 µg C (µg chl $a$)$^{-1}$ d$^{-1}$ with the addition of higher $CO_2$ (Tukey
HSD, $P$ < 0.001) at station SEATS. On the other hand, neither HT (Tukey HSD, $P$ =
0.707) nor HTHC (Tukey HSD, $P$ = 0.057) affected NPP at station D001.

The in situ dark respiration rate at station SEATS was remarkably lower than that

at station D001 regardless of normalizing to volume of seawater (Independent
samples t-test, t = −11.568, df = 4, $P$ < 0.001; Fig. 5a) or chl $a$ (Independent samples
t-test, t = −8.019, df = 4, $P$ = 0.001; Fig. 5b). The higher temperature increased dark
respiration rate from 2.8 $\pm$ 1.2 to 9.3 $\pm$ 0.6 µg C L$^{-1}$ d$^{-1}$ (Tukey HSD, $P$ < 0.001) at
station SEATS while HTHC reduced it to 1.1 $\pm$ 0.2 µg C L$^{-1}$ d$^{-1}$ (Tukey HSD, $P$ =
0.009; Fig. 5c). The higher temperature also promoted dark respiration rate at station
D001, from 16.9 $\pm$ 2.0 to 31.5 $\pm$ 5.1 µg C L$^{-1}$ d$^{-1}$ (Tukey HSD, $P$ = 0.007) but HTHC




did not alter it (Tukey HSD, $P$ = 0.516; Fig. 5c). When it was normalized to chl $a$,
higher temperature still increased dark respiration rate from 12.0 ± 1.8 to 19.7 ± 1.4
μg C (μg chl $a$)$^{-1}$ d$^{-1}$ (Tukey HSD, $P$ = 0.006) while the effect of temperature on
respiration rate at station D001 was insignificant (Tukey HSD, $P$ = 0.891; Fig. 5d).
Compared to LTLC, HTHC did not affect respiration rate at either station (Tukey
HSD, $P$ = 0.131 at station SEATS, $P$ = 0.348 at station D001; Fig. 5d).

The in situ ratio of respiration to photosynthesis was 31.9 ± 3.5% at stations

SEATS, significantly higher than that (24.2 ± 1.3%) at station D001 (Independent
samples t-test, t = 3.537, df = 4, $P$ = 0.0024; Fig. 6a). After seven days incubation in
microcosms, Tukey HSD comparison ($P$ = 0.05) showed that higher temperature did
not affect the ratio of respiration to photosynthesis but HTHC reduced it from 27.8 ±
1.6% to 16.5 ± 1.3% at station SEATS (Fig. 6b). On the contrary, HTHC (38.7 ± 3.1%)
increased the ratio compared to LTLC (27.3 ± 2.4%), with insignificant effect of
temperature alone (29.5 ± 3.3%) at station D001 (Fig. 6b).
**4 Discussion**
**4.1 Effects of increased temperature and $CO_2$ on biomass**

The in situ chl $a$ concentration at station D001 was higher than that at SEATS,

indicating more phytoplankton biomass at station D001. The chl $a$ concentration
decreases with distance from the coast in the NSCS, mainly due to the change of
nutrients (Li et al., 2011). The higher temperature increased chl $a$ concentration at
both stations, which might be attributed to increased active uptake of nutrients at the
elevated temperatures through enhanced enzymatic activities. Algal and
cyanobacterial growth commonly increases with temperature within a suitable range
(Goldman and Carpenter, 1974; Montagnes and Franklin, 2001; Savage et al., 2004;
Boyd et al. 2013) and optimum temperatures for growth of marine phytoplanton are
usually several degrees higher than the environmental temperatures (Thomas et al.,

2012).

On the other hand, the elevated $CO_2$ offset the positive effect of the higher

temperature on chl $a$ at station D001, and even reduced chl $a$ at station SEATS. High
$CO_2$ can sometimes enhance algal photosynthesis and growth, since $CO_2$ in seawater





is suboptimal for full operation of Rubisco enzymes (Wu et al., 2008 and references
therein). On the other hand, positive effects of elevated $CO_2$ can be affected by other
environmental factors. Gao et al. (2012b) demonstrated that rising $CO_2$ could enhance
growth of diatoms at low light intensity, but decrease it at high light intensity. It was
found that rising $CO_2$ concentration lowered the threshold for diatom growth above
which photosynthetic active radiation becomes excessive or stressful, owing to
reduced energy requirements for inorganic carbon acquisition at elevated $CO_2$ (Gao et
al., 2012b). In the present study, the mean daily solar radiation levels during
incubation were 1312 and 1136 μmol photons $m^{-2}$ $s^{-1}$ (Table 1), which were far above
the threshold light intensity reported for diatoms (Gao et al., 2012b). Consequently,
the higher $CO_2$ combined with the high solar radiation in summer of NSCS may have
imposed negative effects on phytoplankton biomass at station SEATS and D001. In
addition, the inhibitory effect of higher $CO_2$ on biomass was more significant at
station SEATS than D001. This can be attributed to the higher sensitivity of
picoplankton to high solar radiation (Li et al., 2011; Wu et al., 2015), which could be
delivered to the interaction of high solar radiation and high $CO_2$. As shown in Li et
al.'s (2011) study, the proportion of picoplankton in phytoplankton assemblages
increased with distance off the coasts. Therefore, the dominant species at station
SEATS are pico- and nano-phytoplankton, but micro-phytoplankton at station D001
(Table 3).

**4.2 Effects of increased levels of temperature and $CO_2$ on primary productivity**

The seawater volume-specific DPP at station D001 was higher than station
SEATS. This should result from both higher chl *a* concentration and chl *a*-specific
DPP at D001. It has been shown that smaller cells exist at SEATS than at D001 (Table
3). Smaller cells have been considered to have larger DPP, according to Laws's model
(1975). The discrepancy between our finding and Laws's model may be due to the
availability of nutrients. Laws's model was based on growth rates obtained from the
same nutrient level. Nevertheless, the nutrient level at station D001 is higher than at
SEATS (Table 3), leading to higher DPP. The higher temperature increased seawater
volume-specific DPP at both stations. This could be attributed to more biomass





produced at the warmer conditions, as indicated by chl *a*. High temperature enhanced
the chl *a*-specific DPP at station SEATS. However, no positive effects of temperature
were found on chl *a*-specific DPP at station D001. The differential effects of
temperature on chl *a*-specific DPP between the stations may be due to the
phytoplankton community composition, since cyanobacteria and/or
pico-phytoplankton have the strongest temperature response in terms of
photosynthetic carbon fixation compared to micro- and nano-phytoplankton
(Andersson et al., 1994). This finding contributes to the explanation of the dominance
of pico-phytoplankton in a warmer ocean (Chen et al., 2014; Montagnes and Franklin,
2001; Hare et al. 2007; Morán et al., 2010). HTHC reduced chl *a*-specific DPP at
station D001, but increased it at station SEATS. High $CO_2$ also reduced chl *a*-specific
DPP in a previous study, which could result from the interaction of high $CO_2$ and high
solar radiation during the summer in the NSCS (Gao et al., 2012b). The reason that
HTHC stimulated chl *a*-specific DPP at station SEATS may be due to a dramatic
decline in chl *a* concentration under the HTHC treatment.
**4.3 Effects of increased temperature and $CO_2$ on respiration**
The dark respiration rate of phytoplankton at station D001 was higher than that at
SEATS, regardless of normalizing to seawater or chl *a*. The respiration rate of algae or
cyanobacteria usually increases with cell size (López-Sandoval et al., 2014). Station
SEATS has more pico-phytoplankton, which would lead to a lower chl *a*-specific dark
respiration rate and then lower seawater volume-specific dark respiration, particularly
when combined with lower chl *a* level. The higher temperature increased seawater
volume-specific dark respiration at both stations, which could be related to increased
chl *a* concentration and/or enhanced respiratory carbon loss at the higher temperature.
The higher temperature also increased chl *a*-specific dark respiration rate at station
SEATS. This is consistent with Butrón et al.'s (2009) study, in which respiration rates
of phytoplankton along Nervioń–Ibaizabal estuary showed a positive correlation with
temperature. Robarts and Zohary (1987) also found that respiration rate of
bloom-forming cyanobacteria was temperature-dependent, with optima over 25 ℃.
HTHC reduced seawater volume-specific dark respiration at both stations, which



should be the consequence of the decreased chl *a* in this treatment. The higher
temperature increased chl *a*-specific dark respiration rate at station SEATS, but there
was no significant difference between HTHC and LTLC, indicating the higher $CO_2$
inhibited the chl *a*-specific dark respiration rate. Similarly, reduced respiration was
found in mesocosm studies (Spilling et al., 2016). In theory, higher $CO_2$ would inhibit
respiratory release of $CO_2$. Nevertheless, enhanced respiration rate at higher $CO_2$
conditions have been found in laboratory-grown diatoms (Wu et al., 2010),
coccolithophores (Jin et al., 2015), mixed phytoplankton assemblages (Jin et al.,
2015), and macroalgae as well (Zou et al., 2011). Such increased respiration has been
attributed to extra energy demand to cope with increased seawater acidity caused by
higher $CO_2$ (Gao and Campbell, 2014; Raven et al., 2014). Therefore, the effect of
increased $CO_2$ on phytoplankton respiration could be due to the combined effects of
$CO_2$ diffusive resistance and seawater acidity stress. Meanwhile, neither higher
temperature alone nor HTHC significantly affected chl *a*-specific dark respiration rate
at station D001. One possible reason could be that larger cells are less sensitive to
$CO_2$ diffusive resistance and acidic stress due to thicker diffusion boundary layers
around the cells. This hypothetical explanation is worthy of future testing.
**4.4 Effects of increased temperature and $CO_2$ on R/P**
Laws's model (1975) has suggested that large phytoplankton cells are likely to
consume a smaller fraction of their biomass to compete with small phytoplankton
cells in terms of the growth rate, considering small cells have higher gross production
rates. Our finding that phytoplankton in station SEATS had a higher R/P than station
D001 confirms Laws's model. It was theorized that autotrophic respiration is more
sensitive to temperature than photosynthesis and the ratio of R/P was predicted to
increase with temperature (Ryan, 1991; Woodwell, 1990; Woodwell et al., 1983).
However, the assumption that plant respiration is highly temperature dependent was
primarily based on short-term (a few minutes or hours) responses of plants to changes
of temperature (Gifford, 1994). In long-term experiments (days or months), the
increase in respiration with temperature tended to disappear due to acclimation
(Gifford, 1995; Jones, 1977; Reich et al., 2016; Slot and Kitajima, 2015; Ziska and





Bunce, 1998). Photosynthetic acclimation to warming is variable (Chalifour and
Juneau, 2011; Hancke and Glud, 2004; Schlüter et al., 2014; Staehr and Birkeland,
2006). However, a general acclimation response to long-term increased temperature is
a rise in optimal temperature of photosynthesis (Gunderson et al., 2010; Kattge and
Knorr, 2007; Staehr and Birkeland, 2006). Such shifts in the temperature response of
photosynthesis and respiration via physiological acclimation can dampen the increase
in R/P at high temperatures, or climate warming would not increase R/P (Reich et al.,
2016). In other words, phytoplankton would down-regulate the high sensitivity of
respiration to temperature, and maintain a relatively consistent net primary production
and hence food web structure in a warming ocean. Although the ratio of R/P did not
vary with increased temperature at both stations in our work either, both
photosynthesis and respiration were enhanced by the higher temperature. The constant
ratio was due to the similar amplitude of increase in photosynthesis and respiration.
The incubation period of 7–8 days might not be enough for phytoplankton to
acclimate to the increased temperature completely, and therefore the stimulated effects
of high temperature on photosynthesis and respiration were still notable. Opposite
effects of HTHC on R/P were detected at station SEATS and D001, negative at
SEATS and positive at D001. This can be attributed to differential responses of
photosynthesis at both stations to HTHC, considering the responses of respiration
were similar.

**5 Conclusions**

This study demonstrates that ocean warming expected to occur by the end of the
century would simulate the DPP and also dark respiration of phytoplankton
assemblages in NSCS, but this positive effect can be damped or offset when ocean
acidification is combined. The responses of phytoplankton assemblages locating
different areas to ocean warming and acidification could be contrasting due to various
phytoplankton compositions and physical and chemical environments. It seems that
phytoplankton assemblages in pelagic areas are more sensitive to ocean warming and
acidification. More exhaustive investigations are needed to obtain an accurate view of
primary production under future ocean environment.





**Acknowledgements**
This study was supported by the national key research programs
2016YFA0601400, National Natural Science Foundation (41430967; 41120164007),
State Oceanic Administration (National Programme on Global Change and Air-Sea
Interaction, GASI-03-01-02-04).

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



Table 1. The daytime (12 h) mean solar radiation (PAR, μmol photons m$^{-2}$ s$^{-1}$) during incubation at stations SEATS and D001.

| SEATS | | D001 | |
|---|---|---|---|
| Date | Solar radiation | Date | Solar radiation |
| 03/08/2012[a] | 1454 | 14/08/2012[a] | 1512 |
| 04/08/2012 | 1304 | 15/08/2012 | 1480 |
| 05/08/2012 | 1146 | 16/08/2012 | 400 |
| 06/08/2012 | 1113 | 17/08/2012 | 111 |
| 07/08/2012 | 927 | 18/08/2012 | 1520 |
| 08/08/2012 | 1592 | 19/08/2012 | 1583 |
| 09/08/2012 | 1582 | 20/08/2012[b] | 1346 |
| 10/08/2012[b] | 1381 | Mean[c] | 1136 |
| Mean[c] | 1312 | | |

[a]The dates for measurements of photosynthetic carbon fixation in situ. [b]The dates for measurements of photosynthetic carbon fixation experiencing temperature and $p$CO$_2$ treatments. [c]Mean represents the average of daytime mean solar radiation over seven or six days microcosm incubation.



Table 2. Parameters of the seawater carbonate system at different incubation
conditions. Measurements and estimation of the parameters were described in
Methods. Data are the means ± SD (n = 3). LTLC, low temperature and low $p$CO$_2$;
HTLC, high temperature and low $p$CO$_2$; HTHC, high temperature and high $p$CO$_2$.
DIC = dissolved inorganic carbon, TA = total alkalinity. Different superscript letters
indicate significant differences between treatments within one station.

| | SEATS | | | D001 | | |
|---|---|---|---|---|---|---|
| | LTLC | HTLC | HTHC | LTLC | HTLC | HTHC |
| Temperature (ºC) | 30.5 ±1.0 | 33.5 ±1.0 | 33.5 ±1.0 | 28.5 ±1.0 | 31.5 ±1.0 | 31.5 ±1.0 |
| pH$_T$ | 8.07 ±0.01 | 8.05 ±0.01 | 7.68 ±0.01 | 8.02 ±0.01 | 8.01 ±0.01 | 7.68 ±0.01 |
| $p$CO$_2$ (µatm) | 390.0 ±19.5 | 390.0 ±19.5 | 1000.0 ±70.0 | 420.0 ±25.2 | 420.0 ±25.2 | 1030.0 ±60.0 |
| DIC (µmol kg$^{-1}$) | 2056.4 ±49.2[a] | 1986.6 ±47.1[a] | 1999.9 ±91.4[a] | 1969 ±67.7[a] | 1896.5 ±64.8[a] | 2039.1 ±69.5[a] |
| HCO$_3^-$ (µmol kg$^{-1}$) | 1758.4 ±47.5[ab] | 1681.2 ±45.4[a] | 1838.4 ±86.5[b] | 1719.2 ±63.7[a] | 1640.9 ±60.7[a] | 1882.3 ±66.4[b] |
| CO$_3^{2-}$ (µmol kg$^{-1}$) | 288.2 ±1.2[b] | 296.2 ±1.2[c] | 138.1 ±3.3[a] | 238.8 ±3.4[b] | 245.3 ±3.5[b] | 131.6 ±1.7[a] |
| CO$_2$ (µmol kg$^{-1}$) | 9.8 ±0.5[a] | 9.1 ±0.5[a] | 23.5 ±1.7[b] | 11.0 ±0.7[a] | 10.3 ±0.7[a] | 25.2 ±1.5[b] |
| TA (µmol kg$^{-1}$) | 2443.5 ±47.9[a] | 2387.7 ±45.8[a] | 2170.7 ±92.0[b] | 2294.2 ±68.6[a] | 2260.0 ±33.9[a] | 2199.4 ±68.5[a] |






Table 3. Physical, chemical, and biological parameters at stations SEATS and D001.
SST: seawater surface temperature; N: $NO_3^- + NO_2$ ($\mu$mol $L^{-1}$); P: $PO_4^{3-}$ ($\mu$mol $L^{-1}$).
Data of nutrients and phytoplankton composition are derived from literatures.

| Station | SST | Salinity | $pH_T$ | N | P | Dominant phytoplankton |
|---|---|---|---|---|---|---|
| SEATS | 28.7 | 32.9 | 8.07 | <0.1[a] | <0.01[b] | Pico- and nano-phytoplankton[c] |
| D001 | 26.8 | 33.5 | 8.03 | >1[d] | >0.1[d] | Micro-phytoplankton[e] |

[a]Du et al. (2013); [b]Wu et al. (2003); [c]Li et al. (2011); [d]Li et al. (2014); [e]Zhang et al. (2014).











**Figure captions**

**Figure 1.** Experimental stations in the northern South China Sea.

**Figure 2.** Chl *a* concentration in situ (a) and after temperature and $pCO_2$ treatments in microcosms (b). The microcosm incubations lasted seven days at station SEATS and six days at station D001.The error bars indicate the standard deviations (n = 3). The different letters above the error bars represent significant ($P < 0.05$) differences between stations in panel (a) and between treatments in panel (b).

**Figure 3.** Daytime primary productivity (DPP) in situ (a, b) and after temperature and $pCO_2$ treatments in microcosms (c, d). The microcosm incubations lasted seven days at station SEATS and six days at station D001. The error bars indicate the standard deviations (n = 3). The different letters above the error bars represent significant ($P < 0.05$) differences between stations in panels (a, b) and between treatments in panels (c, d).

**Figure 4.** Net primary productivity (NPP) in situ (a, b) and after temperature and $pCO_2$ treatments in microcosms (c, d). The microcosm incubations lasted seven days at station SEATS and six days at station D001. The error bars indicate the standard deviations (n = 3). The different letters above the error bars represent significant ($P < 0.05$) differences between stations in panels (a, b) and between treatments in panels (c, d).

**Figure 5.** Dark respiration in situ (a, b) and after temperature and $pCO_2$ treatments in microsoms (c, d). The microcosm incubations lasted seven days at station SEATS and six days at station D001. The error bars indicate the standard deviations (n = 3). The different letters above the error bars represent significant ($P < 0.05$) differences between stations in panels (a, b) and between treatments in panels (c, d).

**Figure 6.** The ratio of respiration to photosynthesis in situ (a, b) and after temperature and $pCO_2$ treatments (c, d) in microcosms. The microcosm incubations lasted seven days at station SEATS and six days at station D001. The error bars indicate the standard deviations (n = 3). The different letters above the error bars represent significant ($P < 0.05$) differences between stations in panels (a, b) and between treatments in panels (c, d).






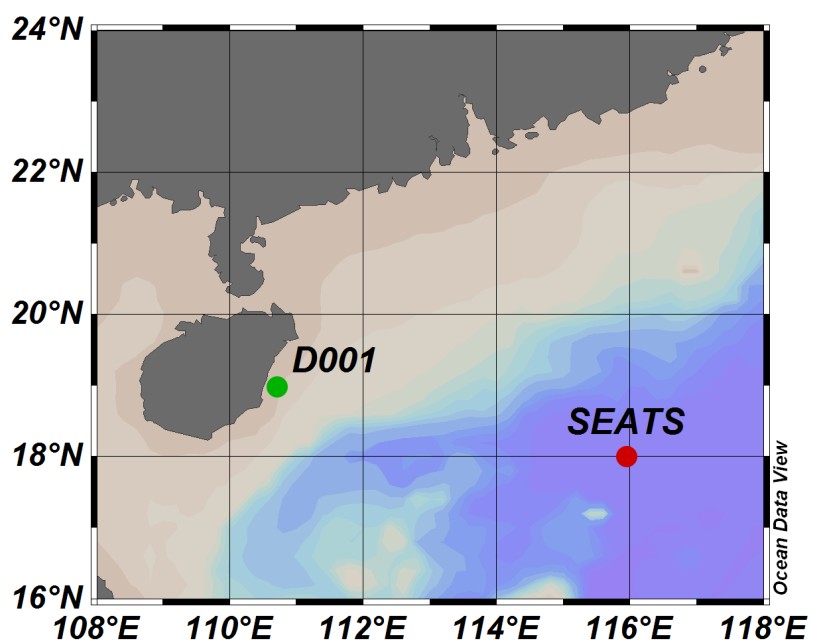


690                                  Figure 1








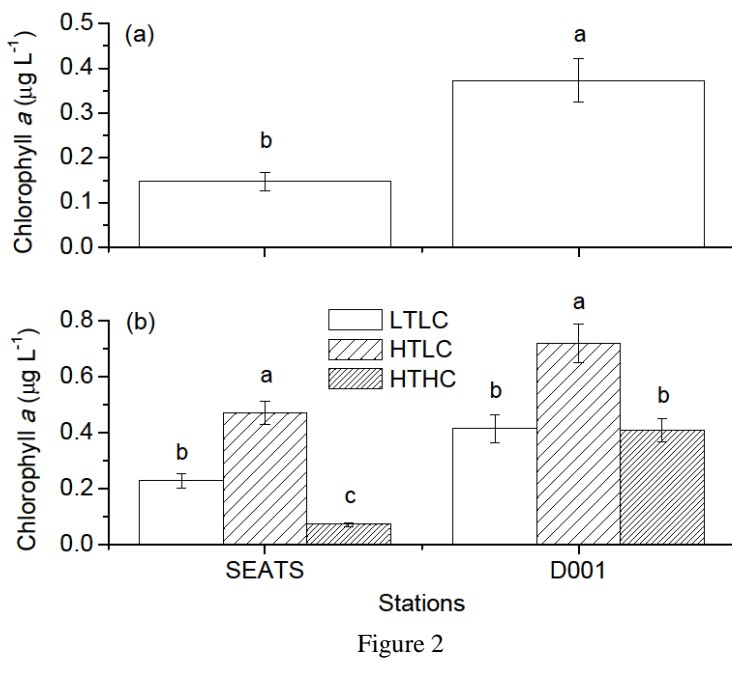

Figure 2





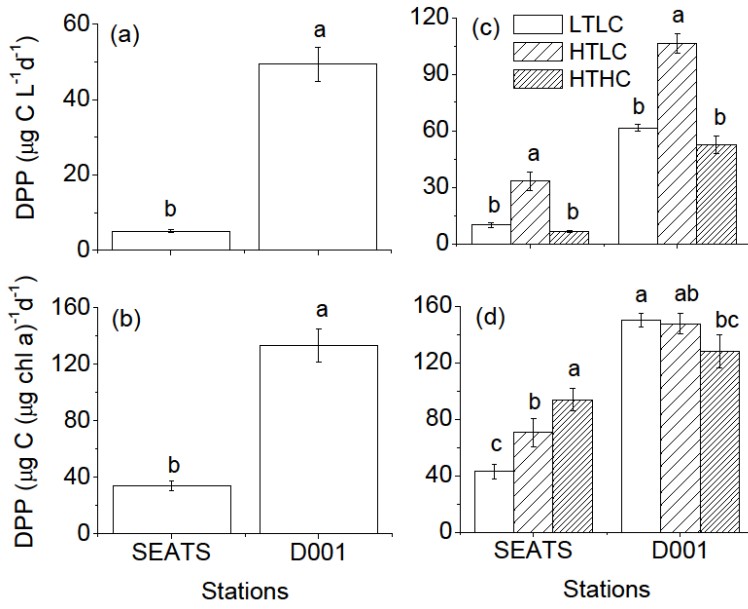

Figure 3





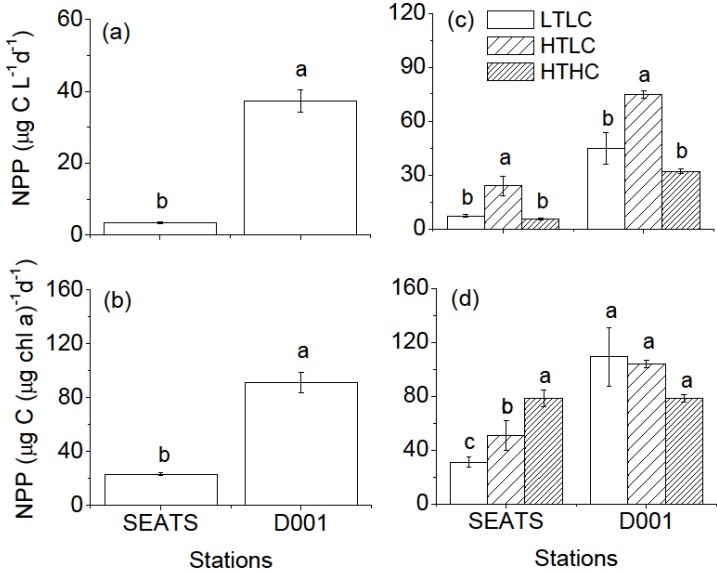

Figure 4





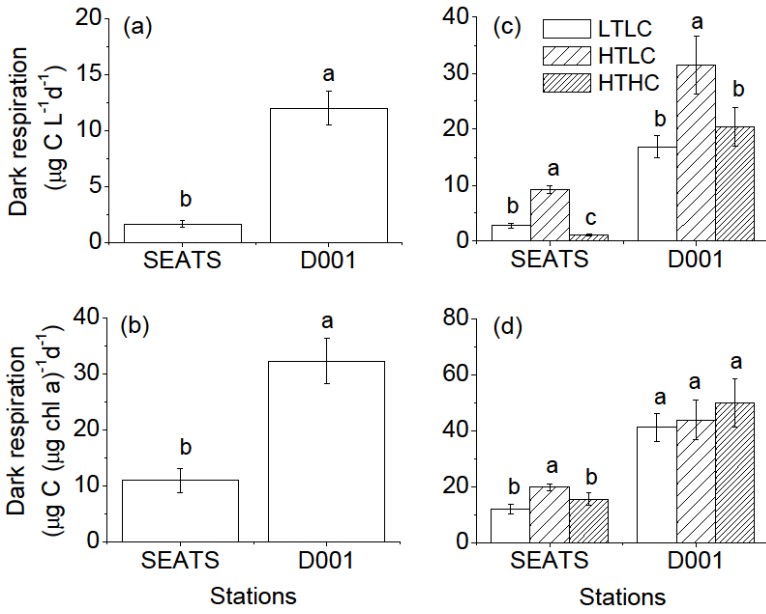

Figure 5



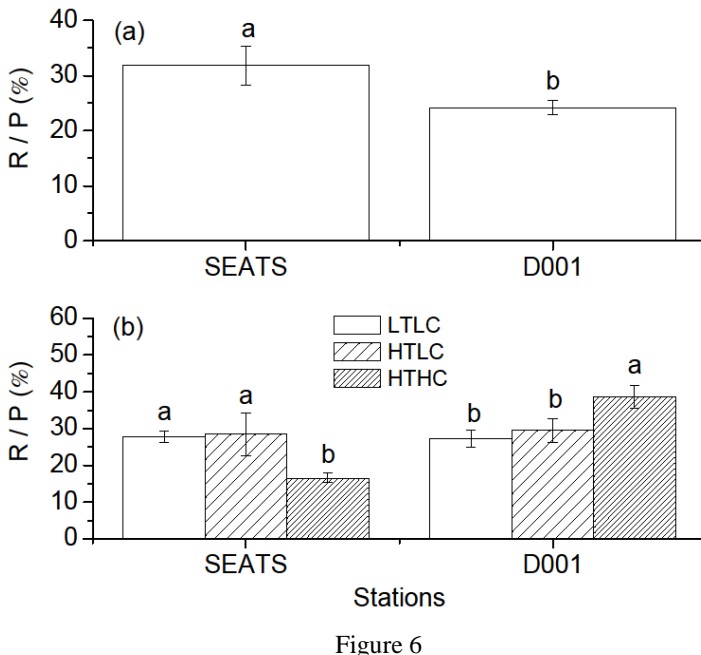

Figure 6