# Peer review of "Combined effects of elevated $p\text{CO}_2$ and temperature on biomass and carbon"

_Biogeosciences, 2016_

## Referee Comment (RC1) · Anonymous Referee #1 · 12 Oct 2016

[Figure]

[Figure]

Figure 1

[Figure]

[Figure]

[Figure]

695     Figure 2

[Figure]

[Figure]

[Figure]

Figure 3

[Figure]

[Figure]

Figure 4

[Figure]

[Figure]

Figure 5

[Figure]

[Figure]

Figure 6

[referee-annotated manuscript omitted]

---

## Referee Comment (RC2) · Anonymous Referee #2 · 18 Nov 2016

The MS "Combined effects of elevated pCO2 and temperature on biomass and carbon fixation of phytoplankton assemblages in the northern South China Sea" by Gao et al. describes one of the first studies trying to understand changes in productivity at two stations (on shore and off shore) the South China Sea to a combination of both enhanced temperature as well as CO2.

The hypothesis to be answered is an important one and we do need high quality datasets to understand potential changes in productivity in a future ocean. Unfortunately I find this manuscript to be written very confusingly with some significant issues on experimental design, data evaluation and interpretation.

Please find my detailed comments below.

[Figure]

Abstract:

Please keep "near shore" and "off-shore" as descriptions of the experimental sites. It becomes very confusing reading SEATS and D001 over and over again.

The flow of the text is constantly interrupted by parentheses – please change.

What does the last sentence mean? "...being more sensitive to these two global change factors" . What does more sensitive mean? In comparison to what? Please clarify.

Introduction:

Line 58 to 67: The authors state different changes in SST increase over time (global SST, South China Sea, global mean rate. At least one of these temperatures is redundant. Also define if you mean South China Sea SST or average temp. Line 73: correct the typo in phytoplankton. (in general please only submit a manuscript after carefully revising it – obvious typos as well as incomplete sentences -see below) should have been revised by at least one co-author!) Line 79: this is not a sentence. – also this non-sentence needs a reference. Line 82/83: Add a reference. Line 85: Define RCP scenario Line 95: change the wording "problem" Line 98: Gao et al 2012a was certainly not the first suggesting energy and metabolite allocation from the down-regulation of the CCM. Despite -to my knowledge the cited study (being a great study) did not investigate any CCM parameters. Line 101: The cited papers represent only a minor fraction of papers with a "neutral" $CO_2$ response – maybe use a review paper as citation instead adding "and references therein" Line 125: what do the authors mean with "assimilation number"?

Methods: Some general remarks: Why did the authors not shield the incubations from the very high light intensity? Intensities of >1000 $\mu$mol photons m-2s-1 as they correctly stated are pretty high and most (all) other studies used much lower light intensities. And phytoplankton change their vertical position over the day!

[Figure]

Did the authors monitor the temperature in the incubation continuously? The tank can easily heat up several degrees over the day if not monitored carefully. Please add this information to the MS.

For future reference, it is best practice to measure at least DIC or TA additional to pH for these kinds of experiments. I know several reviewers who would not accept this MS just based on the "sloppy" characterization of the carbonate chemistry.

Detailed comments:

It is unclear when the DDT was actually incubated. When was the sample taken and when was the 14C added. Was the experiment run every day or was it run once at the end of the incubation.

Please revise the method section of the experimental setup and the corresponding measurements in order to understand the timeline of the measurements during the incubation.

Urgently needed additional information: Nutrient concentrations (specific values and not > or < ( see Table 3) ) prior to the acclimation start as well as nutrient concentrations at the end of the incubation.

Regarding the statistical analysis: I assume that the authors had a maximum of three replicates. The authors state that the data "were conformed to a normal distribution". This seems to me pretty much impossible. How can n=3 be considered a normal distribution? Did the authors verify the outcome of the Shapiro-Wilk test? Please clarify!

Results:

Line 208-215: Most of the results listed here are basic carbonate chemistry responses – please shorten this section.

The authors compare the initial chl a concentration of both stations. As the authors

know – the chlorophyll concentration is changing daily – even hourly, can be different 500 meter next to the sampling spot and obviously will vary with season. This whole section including the discussion does not mean anything if you don't look at long-term changes and differences. Please revise!

Line 255 -276: I feel that without the information on nutrient concentration in the different acclimations any data obtained are oblivious. Please revise if nutrient data are available.

Discussion: Line 284-288: The authors state that phytoplankton growth commonly increases with temperature This statement is simply wrong as phytoplankton which growth at its optimal temperature will be heat stressed at even higher temperature. Please revise. Also – the authors did not test a full temperature growth response curve for the phytoplankton in the North China Sea.

Line 293: Please change the citation (Wu et al. 2008) to a more original work. I also feel that the citation culture in this manuscript should be improved.

In the conclusion the authors state that the study demonstrates that ocean warming would stimulate DPP and that the effect can be dampened by OA. This would be an important result – but I feel that the data quantity and quality does not support this very general conclusion. The authors also disqualified their results with their own discussion in line 378-402 when they talk about the shortcomings of short time vs. long time acclimations. This manuscript is a short time incubation and should be discussed as is including the potential shortcomings.

---

## Author Comment (AC1) · 20 Dec 2016

[revised manuscript text omitted]

                      Figure 1

[Figure]

[Figure]

[Figure]

                          Figure 2

[Figure]

[Figure]

Figure 3

[Figure]

[Figure]

[Figure]

Figure 4

[Figure]

[Figure]

Figure 5

[Figure]

[Figure]

Figure 6

---

## Author Comment (AC2) · 20 Dec 2016

Anonymous Referee #2 The MS "Combined effects of elevated pCO2 and temperature on biomass and carbon fixation of phytoplankton assemblages in the northern South China Sea" by Gao et al. describes one of the first studies trying to understand changes in productivity at two stations (on shore and off shore) the South China Sea to a combination of both enhanced temperature as well as CO2.

The hypothesis to be answered is an important one and we do need high quality datasets to understand potential changes in productivity in a future ocean. Unfortunately I find this manuscript to be written very confusingly with some significant issues on experimental design, data evaluation and interpretation. Please find my detailed comments below.

Response: We have carefully examined the comments raised by the reviewer, and believe our manuscript has been largely improved now.

Abstract: Please keep "near shore" and "off-shore" as descriptions of the experimental sites. It becomes very confusing reading SEATS and D001 over and over again.

Response: Corrected.

The flow of the text is constantly interrupted by parentheses – please change.

Response: Corrected. Please see lines 32-40.

What does the last sentence mean? "...being more sensitive to these two global change factors". What does more sensitive mean? In comparison to what? Please clarify.

Response: More sensitive means being more easily affected by these two global change factors in comparison to the coastal phytoplankton communities. It has been clarified to "with the pelagic phytoplankton communities being more easily affected by these two global change factors in terms of carbon fixation and respiration compared to the coastal phytoplankton communities" at lines 54-56.

Introduction: Line 58 to 67: The authors state different changes in SST increase over time (global SST, South China Sea, global mean rate. At least one of these temperatures is redundant. Also define if you mean South China Sea SST or average temp.

Response: The global mean rate has been removed and the warming rate for South China Sea is now what is meant by SST. Therefore, that sentence has been changed to "From 1979 to 2012, the mean global sea surface temperature increased at a rate of $\sim$0.12°C per decade (IPCC, 2013). Particularly, the warming rate in the surface of South China Sea ($\sim$0.26 per decade) from 1982 to 2004 (Fang et al., 2006) appears to be about 2 times faster" at lines 63-68.

Line 73: correct the typo in phytoplankton. (in general please only submit a manuscript after carefully revising it – obvious typos as well as incomplete sentences -see below) should have been revised by at least one co-author!)

Response: We truly apologize for that. We have double checked the text and corrected phytoplanton to phytoplankton at line 316.

Line 79: this is not a sentence. – also this non-sentence needs a reference.

Response: It has been corrected to "Increased atmospheric CO2 is responsible for global warming, and the oceans have absorbed approximately 30% of the emitted anthropogenic carbon dioxide (IPCC, 2013)" at lines 85-87.

Line 82/83: Add a reference.

Response: The following reference has been added at line 89.

Orr J C, Fabry V J, Aumont O, et al. Anthropogenic ocean acidification over the twenty-first century and its impact on calcifying organisms. Nature, 2005, 437: 681-686.

Line 85: Define RCP scenario

Response: It has been clarified to "By 2100, the projected decline in global-mean surface pH is approximately 0.065 (RCP 2.6) to 0.31 (RCP 8.5) (IPCC, 2013)." at lines 90-92.

Line 95: change the wording "problem"

Response: It has been changed to "challenges" at line 101.

Line 98: Gao et al 2012a was certainly not the first suggesting energy and metabolite allocation from the down-regulation of the CCM. Despite -to my knowledge the cited study (being a great study) did not investigate any CCM parameters.

Response: That reference has been changed to Wu et al. (2010), which shows in- creased Km (DIC concentration required for half-maximal photosynthetic rate) by different techniques (PAM and C14), down-regulated CCM, and enhanced growth in Phaeodactylum tricornutum grown at the higher pCO2 condition.

Wu Y, Gao K, Riebesell U. CO2-induced seawater acidification affects physiological performance of the marine diatom Phaeodactylum tricornutum. Biogeosciences, 2010, 7: 2915–2923.

Line 101: The cited papers represent only a minor fraction of papers with a "neutral" CO2 response – maybe use a review paper as citation instead adding "and references therein".

Response: It has been changed to "(Gao et al., 2012 and references therein)" at line 109.

Gao K, Helbling E W, Häder D P, Hutchins D A. Responses of marine primary producers to interactions between ocean acidification, solar radiation, and warming. Marine Ecology Progress Series, 2012, 470: 167-189.

Line 125: what do the authors mean with "assimilation number"?

Response: It is the carbon fixed per chl a per time, now it has been changed to "photosynthetic carbon fixation rate" at line 132.

Methods: Some general remarks: Why did the authors not shield the incubations from the very high light intensity? Intensities of >1000 _mol photons m-2s-1 as they correctly stated are pretty high and most (all) other studies used much lower light intensities. And phytoplankton change their vertical position over the day!

Response: The reviewer raised an important point. In the microcosms, practically, due to the shielding of the cover, water-jacket and the depth of the water, phytoplankton assemblages could be exposed to 81-91% and 64-91% of full sunlight at the off-shore and near-shore stations respectively. Since we aimed to investigate the impacts on surface phytoplankton assemblages, and provide comparable data with the previous study in the SCS (Gao et al., 2012), we decided not to shield the microcosms. These explanations have been added to lines 151-158. As reported previously (Gao et al. 2012), diatoms' response to OA depends on light levels. We have discussed this context at lines 324-338.

Gao K, Xu J, Gao G, Li Y, Hutchins D A, Huang B, Wang L, Zheng Y, Jin P, Cai X, Häder D P, Li W, Xu K, Liu N, Riebesell U. Rising $CO_2$ and increased light exposure synergistically reduce marine primary productivity. Nature Climate Change, 2012, 2: 519-523.

Did the authors monitor the temperature in the incubation continuously? The tank can easily heat up several degrees over the day if not monitored carefully. Please add this information to the MS.

Response: It has been clarified to "Microcosm temperature was controlled and monitored via circulating coolers (CTP-3000, EYELA, Japan) with a variation of $\pm$ 1.0oC" at lines 162-164.

For future reference, it is best practice to measure at least DIC or TA additional to pH for these kinds of experiments. I know several reviewers who would not accept this MS just based on the "sloppy" characterization of the carbonate chemistry.

Response: We would like to thank the reviewer for this advice. It would be helpful to double check the chemistry stability. In this work, we measured pH and $pCO_2$ in the microcosms and other carbonate system parameters were derived via CO2SYS (Pierrot et al., 2006), using the equilibrium constants of K1 and K2 for carbonic acid dissociation (Roy et al., 1993).

Pierrot D, Lewis E, and Wallace D W R. MS Excel program developed for $CO_2$ system calculations, ORNL/CDIAC-105a. Carbon Dioxide Information Analysis Center, Oak Ridge National Laboratory, US Department of Energy, Oak Ridge, Tennessee, 2006. 2006. Roy RN, Roy LN, Vogel KM, Porter-Moore C, Pearson T, Good CE, et al. The dis-

sociation constants of carbonic acid in seawater at salinities 5 to 45 and temperatures 0 to 45oC. Marine Chemistry, 1993, 44: 249-267.

Detailed comments: It is unclear when the DDT was actually incubated. When was the sample taken and when was the 14C added. Was the experiment run every day or was it run once at the end of the incubation. Please revise the method section of the experimental setup and the corresponding measurements in order to understand the timeline of the measurements during the incubation.

Response: Detailed information has been added to the Methods section. It reads "The concentrations of chl a in situ and in microcosms were measured at the beginning and end of the experiment, respectively" at line 186-188; "Seawater samples taken from each microcosm at the end of the experiment were dispensed into 50 mL quartz tubes, inoculated with 5 $\mu$Ci (0.185 MBq) NaH14CO3 (ICN Radiochemicals, USA) and then incubated for 12 h (from 6:00 a.m. to 6:00 p.m.) and 24 h (from 6:00 a.m. to 6:00 a.m. next day) under natural light and day-night conditions" at lines 190-194; "The primary productivity and dark respiration in situ were measured at the beginning of the experiment." at lines 206-208.

Urgently needed additional information: Nutrient concentrations (specific values and not > or < (see Table 3) ) prior to the acclimation start as well as nutrient concentrations at the end of the incubation.

Response: We realize that nutrient concentration is an important parameter. Unfortunately, we did not measure it during this ship-board experiment. Instead, we used reported data of the nutrients in the same regions for the reference (Table 3). We will measure nutrient concentrations in our future work by all means.

Regarding the statistical analysis: I assume that the authors had a maximum of three replicates. The authors state that the data "were conformed to a normal distribution". This seems to me pretty much impossible. How can n=3 be considered a normal distribution? Did the authors verify the outcome of the Shapiro-Wilk test? Please clarify!

Response: Yes, we had triplicate microcosms for each treatment. We realize that triplicates are the minimum for statistical analysis, but we did not have more replicates due to the breakdown of some microcosms during transport. We double checked the outcome of the Shapiro-Wilk tests and found all data sets were conformed to a normal distribution except the daily primary productivity (P = 0.034) and dark respiration (P = 0.034) under HTLC at the near-shore station. Consequently, we have changed the text to "The data from each treatment conformed to a normal distribution (Shapiro-Wilk, P > 0.05) except the DPP (P = 0.034) and dark respiration (P = 0.034) under HTLC at the near-shore station" at lines 211-213. We are happy to supply the outcome of statistical analysis if necessary.

Results: Line 208-215: Most of the results listed here are basic carbonate chemistry responses – please shorten this section.

Response: This paragraph describes changes of carbonate chemistry under three treatments at two stations using only eight lines. To shorten this section will affect clarification of the results and therefore we hope the current text could be sustained.

The authors compare the initial chl a concentration of both stations. As the authors know – the chlorophyll concentration is changing daily – even hourly, can be different 500 meter next to the sampling spot and obviously will vary with season. This whole section including the discussion does not mean anything if you don't look at long-term changes and differences. Please revise!

Response: We agree with the reviewer on the dynamics of chl a concentration, which could be affected by both physical and chemical forcings. The two stations, one is pelagic and highly oligotrophic, and the other station is near the coastal up-welling. Therefore, naturally, mean chl a concentration would be very different, despite the inevitable variability the reviewer points out. We have revised this part, trying to focus on responses of different regions or water bodies to changed pH, pCO2 and temperature,

at lines 238-247 and 307-319.

Line 255 -276: I feel that without the information on nutrient concentration in the different acclimations any data obtained are oblivious. Please revise if nutrient data are available.

Response: Yes, we totally agree with the reviewer that the conditions of nutrients are important. Since we compared the regional (different stations) response to pH and temp changes, and we had control for each station, it is, we believe, informative in terms of the phytoplankton assemblages' response to climate change in different areas. We did not measure the nutrients in the seawater but we know the general ranges of nutrients for both stations based on the references, as mentioned in the above response. We will measure nutrient concentrations in our future work by all means.

Discussion: Line 284-288: The authors state that phytoplankton growth commonly increases with temperature This statement is simply wrong as phytoplankton which growth at its optimal temperature will be heat stressed at even higher temperature. Please revise. Also – the authors did not test a full temperature growth response curve for the phytoplankton in the North China Sea.

Response: We have realized that the wordings are confusing. We have revised, and now it reads "Algal and cyanobacterial growth commonly increases with temperature within a suitable range and then decreases after the optimal temperature point/range (Goldman and Carpenter, 1974; Montagnes and Franklin, 2001; Savage et al., 2004; Boyd et al. 2013) and optimum temperatures for growth of marine phytoplankton are usually several degrees higher than the environmental temperatures (Thomas et al., 2012), which could explained the increase chl a level of phytoplankton grown at the higher temperature in the present study." at lines 312-319. We admit that more information would have been obtained if a full temperature growth response curve had been conducted. We did not do it because our aim was only to examine the effect of projected temperature and pCO2 by the end of this century on primary productivity in

SCS.

Line 293: Please change the citation (Wu et al. 2008) to a more original work. I also feel that the citation culture in this manuscript should be improved.

Response: This citation has been replaced with Giordano et al. (2005). We have tried to improve the citation culture by citing more original and more related literature. For instance, the citation Gao et al 2012a at line 104 was replaced with Wu et al., 2010, the citation Wu et al., 2008 at line 199 was replaced with Giordano et al. (2005), and the citation (Gao et al., 2007) at line 324 was removed.

Giordano M, Beardall J, Raven J A. $CO_2$ concentrating mechanisms in algae: mechanisms, environmental modulation, and evolution. Annual Review of Plant Biology, 2005, 56: 99-131. Wu Y, Gao K, Riebesell U. $CO_2$-induced seawater acidification affects physiological performance of the marine diatom Phaeodactylum tricornutum. Biogeosciences, 2010, 7: 2915–2923.

In the conclusion the authors state that the study demonstrates that ocean warming would stimulate DPP and that the effect can be dampened by OA. This would be an important result – but I feel that the data quantity and quality does not support this very general conclusion. The authors also disqualified their results with their own discussion in line 378-402 when they talk about the shortcomings of short time vs. long time acclimations. This manuscript is a short time incubation and should be discussed as is including the potential shortcomings.

Response: We have to admit that ship-board tests are hard to extend longer than the ship-cruising period. And we take the advice from the reviewer and revised the conclusion section. It reads now: "This study demonstrates that a short-term rise of SST appeared to simulate the DPP and dark respiration of phytoplankton assemblages in the NSCS. However, this positive effect was dampened or offset when warming and ocean acidification treatments were combined. The regional responses of phytoplankton assemblages at the two stations to ocean warming and acidification may differ due to differences in physical and chemical environment as well as phytoplankton community structure. The combined treatment of warming and acidification reduced biomass and dark respiration rate at the off-shore, but did not affect them at the near-shore station. Ecologically and geographically, our data implies differential responses of primary production to ocean climate change. This short-term experiment suggests the need to determine whether similar effects may occur over the longer timescales of future anthropogenic change." at lines 454-465.

Please also note the supplement to this comment:
http://www.biogeosciences-discuss.net/bg-2016-403/bg-2016-403-AC2-supplement.pdf

**Supplement:**

**Combined effects of elevated $p$CO$_2$ and temperature on biomass and carbon**

**fixation of phytoplankton assemblages in the northern South China Sea**

Guang Gao[1,2], Peng Jin[1], Nana Liu[1], Futian Li[1], Shanying Tong[1], David A. Hutchins[3], and Kunshan Gao[1*]

[1] *State Key Laboratory of Marine Environmental Science, Xiamen University, Xiamen*

*361005, China*

[2] *Marine Resources Development Institute of Jiangsu, Huaihai Institute of Technology,*

*Lianyungang 222005, China*

[3] *Marine Environmental Biology, Department of Biological Sciences, University of*

*Southern California, 3616 Trousdale Parkway, Los Angeles, California 90089, USA*

[*] Correspondence: KunshanGao, Fax: 86-592-2187963, e-mail: ksgao@xmu.edu.cn.

**Abstract**

The individual influences of ocean warming and acidification on marine organisms have been investigated intensively, but studies regarding the combined effects of both global change variables on natural marine phytoplankton assemblages are still scarce. Even fewer studies have addressed possible differences in the responses of phytoplankton communities in pelagic and coastal zones to ocean warming and acidification. We conducted shipboard microcosm experiments at both off-shore  and near-shore  stations in the northern South China Sea (NSCS) under three treatments, low temperature  and low $pCO_2$ ( (LTLC, ambient temperature and ambient $pCO_2$), high temperature  and low $pCO_2$ ( (HTLC, ambient temperature + 3 °C and ambient $pCO_2$), and high temperature  and high $pCO_2$ ( (HTHC, ambient temperature + 3 °C and ambient $pCO_2$ + 610 µatm). Biomass of phytoplankton at both stations were enhanced by HT. HTHC did not affect phytoplankton biomass at near-shore station  but decreased it at the off-shore station . At this off-shore station HT alone increased daily primary productivity (DPP, µg C (µg chl $a$)$^{-1}$ d$^{-1}$) by ~64%, and by ~117% when higher $pCO_2$ was added. In contrast, HT alone did not affect DPP and HTHC reduced it by ~15% at the near-shore station . HT enhanced the dark respiration rate (µg C (µg chl $a$)$^{-1}$ d$^{-1}$) by 64% at the near-shore station , but had no significant effect at the near-shore station , and did not change the ratio of respiration to photosynthesis at either station. HTHC did not affect dark respiration rate (µg C (µg chl $a$)$^{-1}$ d$^{-1}$) at either station compared to LTLC. HTHC reduced the respiration to photosynthesis ratio by ~41% at the off-shore station  but increased it ~42% at the near-shore station . Overall, our findings indicate that responses of coastal and offshore phytoplankton assemblages in NSCS to ocean warming and acidification are contrasting, with the pelagic phytoplankton communities being more easily affected by these two global change factors in terms of carbon fixation and respiration compared to the coastal phytoplankton communities  .

**Key words:** ocean acidification, ocean warming, photosynthesis, primary productivity, respiration, South China Sea

**1 Introduction**

From 1979 to 2012, the mean global sea surface temperature (SST) increased at a rate of ~0.12 ℃ per decade (IPCC, 2013). Particularly, the warming rate in the surface of South China Sea (~0.26 ℃ per decade) from 1982 to 2004 (Fang et al., 2006) appears to be about  2 times faster ~~than the global ocean mean rate (~0.14 per decade) from 1960 to 1990 (Casey and Cornillon, 2001; Fang et al., 2006)~~. It is extremely likely that more than half of the observed increase in global average surface temperature from 1951 to 2010 was caused by the anthropogenic increase in greenhouse gas concentrations and other anthropogenic forcings. Ocean warming is projected to rise approximately by 0.6 ℃ (Representative Concentration Pathway (RCP) 2.6) to 2.0 ℃ (RCP 8.5), in the upper 100 m of the water column by the end of the 21$^{st}$ century (IPCC, 2013).

[revised manuscript text omitted]

acetone at -20 ℃ for 24 h. Chl *a* concentration was determined with a fluorometer (Trilogy, Turner Designs, USA), following the protocol of Welschmeyer (1994). The concentrations of chl *a* in situ and in microcosms were measured at the beginning and end of the experiment, respectively.

**2.5 Primary productivity and dark respiration**

Seawater samples taken from each microcosm at the end of the experiment were dispensed into 50 mL quartz tubes, inoculated with 5 μCi (0.185 MBq) NaH$^{14}$CO$_3$

(ICN Radiochemicals, USA) and then incubated for 12 h (from 6:00 a.m. to 6:00 p.m.)

and 24 h (from 6:00 a.m. to 6:00 a.m. next day) under natural light and day-night conditions. The incubation temperature of every treatment was the same as the corresponding microcosm treatment. After the incubation, the cells were filtered onto a Whatman GF∕F glass fiber filter (25 mm), which was immediately frozen and stored at -20$^{o}$C for later analysis. In the laboratory, each frozen filter was placed into a 20

mL scintillation vial, exposed to HCl fumes overnight, and dried (55 $^{o}$C, 6 h) to expel non-fixed $^{14}$C (Gao et al., 2007). Then 3 mL scintillation cocktail (Perkin Elmer®)

was added to each vial and incorporated radioactivity was counted by a liquid scintillation counting (LS 6500, Beckman Coulter, USA). The daytime primary productivity (DPP) was defined as the amount of carbon fixation during 12 h incubation. The dark respiration was defined as the difference in amount of carbon fixation between 12 h and 24 h. Carbon fixation over 24 h was taken as daily net primary productivity (NPP). Ratio of respiration to photosynthesis (R/P) was expressed as that of respiratory carbon loss to daytime carbon fixation. The primary productivity and dark respiration in situ were measured at the beginning of the experiment.

**2.6 Statistical analyses**

Results were expressed as means of replicates ± standard deviation. Data were analyzed using the software SPSS v.21. The data from each treatment conformed to a normal distribution (Shapiro-Wilk, $P > 0.05$) except the DPP ($P = 0.034$) and dark respiration ($P = 0.034$) under HTLC at the near-shore station, and the variances could be considered equal (Levene's test, $P > 0.05$). One-way ANOVAs were conducted to assess the significant differences in carbonate chemistry parameters, chl $a$, DPP, NPP, dark respiration, ratio of dark respiration to photosynthesis between three treatments. Tukey HSD was conducted for post hoc investigation. Independent samples t-tests were conducted to compare in situ chl $a$, DPP, NPP, dark respiration, and ratio of dark respiration to photosynthesis between both stations. The threshold value for determining statistical significance was $P < 0.05$.

**3 Results**

The incident solar radiation during the experiment was recorded (Table 1). The daytime (12 h) mean solar radiation ranged from 927 to 1592 μmol photons $m^{-2}$ $s^{-1}$ at the off-shore station SEATS while the lowest solar radiance was only 111 μmol photons $m^{-2}$ $s^{-1}$, with the highest of 1583 μmol photons $m^{-2}$ $s^{-1}$, at the near-shore station D001. The average of daytime mean solar radiation over the microcosm incubation at the off-shore station SEATS was 15.52% higher than that at the near-shore station D001.

The changes of the seawater carbonate system under different conditions are shown in Table 2. At the off-shore station SEATS, an increase of 3 $^{o}$C in temperature (HTLC) did not alter carbonate parameters except leading to enhanced $CO_3^{2-}$ (Tukey HSD, $P = 0.009$). HTHC resulted in a significant decrease in $CO_3^{2-}$ (Tukey HSD, $P <$

0.001) and TA (Tukey HSD, $P$ = 0.016) but an increase in $CO_2$ (Tukey HSD, $P$ < 0.001) compared with LTLC. The effect of temperature on carbonate parameters at the near-shore station  were similar to the off-shore station , while HTHC increased $HCO_3^-$ (Tukey HSD, $P$ = 0.046) and did not affect TA (Tukey HSD, $P$ = 0.203).

The in situ chl $a$ levels at the near-shore station and the off-shore station were 0.37 ± 0.05 μg $L^{-1}$ and 0.15 ± 0.02 μg $L^{-1}$, respectively. After seven days incubation in the microcosms at the off-shore station  Tukey comparison *(P* = 0.05) showed that higher temperature (0.46 ± 0.04 μg $L^{-1}$) increased chl $a$ compared with LTLC (0.23 ± 0.03 μg $L^{-1}$) while HTHC (0.07 ± 0.01 μg $L^{-1}$) reduced it (Fig. 2b). The higher temperature (0.72 ± 0.07 μg $L^{-1}$) also increased chl $a$ compared with LTLC (0.41 ± 0.05 μg $L^{-1}$) at the near-shore station  (Tukey HSD, $P$ = 0.001; Fig. 1b), but with no effect of HTLC (0.41 ± 0.04 μg $L^{-1}$) (Tukey HSD, $P$ = 0.988).

The in situ DPP at the near-shore station  (49.4 ± 4.5 μg C $L^{-1}$ $d^{-1}$ or 133.4 ± 12.1 μg C (μg chl $a$)$^{-1}$ $d^{-1}$) was dramatically higher than that at the off-shore station  (5.1 ± 0.5 μg C $L^{-1}$ $d^{-1}$ or 34.1 ± 3.1 μg C (μg chl $a$)$^{-1}$ $d^{-1}$), whether normalized to volume of seawater (Independent samples t-test, t = −17.056, df = 4, $P$ < 0.001; Fig. 3a) or chl $a$ (Independent samples t-test, t = −13.786, df = 4, $P$ < 0.001; Fig. 3b). After seven days incubation in microcosms, the DPP normalized to volume of seawater under HTLC (33.2 ± 4.8 μg C $L^{-1}$ $d^{-1}$) at the off-shore station  was significantly higher than that under LTLC (9.9 ± 1.2 μg C $L^{-1}$ $d^{-1}$) and HTHC (6.6 ± 0.6 μg C $L^{-1}$ $d^{-1}$) (Tukey HSD, $P$ < 0.001) while the difference between LTLC and HTHC was insignificant (Tukey HSD, $P$ = 0.380; Fig. 3c). The pattern at the near-shore station  was similar to the off-shore station (Fig. 2c). When DPP was normalized to chl $a$, the higher temperature increased primary productivity from 43.2 ± 5.1 to 70.7 ± 10.1 μg C (μg chl $a$)$^{-1}$ $d^{-1}$ (Tukey HSD, $P$ = 0.014) and further to 93.9 ± 8.1 μg C (μg chl $a$)$^{-1}$ $d^{-1}$ (Tukey HSD, $P$ < 0.001) when higher $CO_2$ was combined at the off-shore station  (Fig. 2d). In contrast, temperature did not affect DPP (Tukey HSD, $P$ =

0.0924) and HTHC reduced it from 150.3 $\pm$ 4.9 to 128.0 $\pm$ 11.5 µg C (µg chl $a$)$^{-1}$ d$^{-1}$

(Tukey HSD, $P$ = 0.039) at the near-shore station  (Fig. 3d).

The in situ NPP at the off-shore and the near-shore  were 3.5 $\pm$ 0.1 µg C L$^{-1}$ d$^{-1}$ (23.2 $\pm$ 1.0 µg C (µg chl $a$)$^{-1}$ d$^{-1}$) and 37.4 $\pm$ 3.1 µg C L$^{-1}$ d$^{-1}$ (91.2 $\pm$ 7.5 µg C (µg chl $a$)$^{-1}$ d$^{-1}$) respectively, which indicates that the near-shore station  has higher NPP, irrespective of normalizing to volume of seawater (Independent samples t-test, t = −18.998, df = 4, $P$ < 0.001; Fig. 4a) or chl $a$ (Independent samples t-test, t = −15.511, df = 4, $P$ < 0.001; Fig. 4b). After a seven-day incubation in the microcosms, the higher temperature increased NPP to 23.9 $\pm$ 5.3 µg C L$^{-1}$ d$^{-1}$ (Tukey HSD, $P$ = 0.001) while HTHC (5.5 $\pm$ 0.4 µg C L$^{-1}$ d$^{-1}$) did not change it (Tukey HSD, $P$ = 0.793) compared with LTLC (7.2 $\pm$ 0.8 µg C L$^{-1}$ d$^{-1}$). The effects of temperature and $CO_2$ on NPP at the near-shore station  were similar to that at the off-shore station . When NPP was normalized to chl $a$, the higher temperature increased NPP from 31.1 $\pm$ 3.5 to 50.9 $\pm$ 11.3 (Tukey HSD, $P$ = 0.044) and further to 78.3 $\pm$ 5.9 µg C (µg chl $a$)$^{-1}$ d$^{-1}$ with the addition of higher $CO_2$ (Tukey HSD, $P$ < 0.001) at the off-shore station . On the other hand, neither HT (Tukey HSD, $P$ = 0.707) nor HTHC (Tukey HSD, $P$ = 0.057) affected NPP at the near-shore station .

The in situ dark respiration rate at the off-shore station  was remarkably lower than that at the near-shore station  regardless of normalizing to volume of seawater (Independent samples t-test, t = −11.568, df = 4, $P$ < 0.001; Fig. 5a) or chl $a$ (Independent samples t-test, t = −8.019, df = 4, $P$ = 0.001; Fig. 5b). The higher temperature increased dark respiration rate from 2.8 $\pm$ 1.2 to 9.3 $\pm$ 0.6 µg C L$^{-1}$ d$^{-1}$ (Tukey HSD, $P$ < 0.001) at the off-shore station  while HTHC reduced it to 1.1 $\pm$ 0.2 µg C L$^{-1}$ d$^{-1}$ (Tukey HSD, $P$ = 0.009; Fig. 5c). The higher temperature also promoted dark respiration rate at the near-shore station , from 16.9 $\pm$ 2.0 to 31.5 $\pm$ 5.1 µg C L$^{-1}$ d$^{-1}$ (Tukey HSD, $P$ = 0.007) but HTHC did not alter it (Tukey HSD, $P$ = 0.516; Fig. 5c). When it was normalized to chl $a$, higher temperature still increased dark respiration rate from 12.0 $\pm$ 1.8 to 19.7 $\pm$ 1.4 µg C (µg chl $a$)$^{-1}$ d$^{-1}$ (Tukey HSD, $P$ = 0.006) while the effect of temperature on respiration rate at the near-shore station

 was insignificant (Tukey HSD, $P = 0.891$; Fig. 5d). Compared to LTLC, HTHC did not affect respiration rate at either station (Tukey HSD, $P = 0.131$ at the off-shore station , $P = 0.348$ at near-shore station ; Fig. 5d).

The in situ ratio of respiration to photosynthesis was $31.9 \pm 3.5\%$ at the off-shore station  significantly higher than that ($24.2 \pm 1.3\%$) at the near-shore station  (Independent samples t-test, t = 3.537, df = 4, $P = 0.0024$; Fig. 6a). After seven days incubation in microcosms, Tukey HSD comparison ($P = 0.05$) showed that higher temperature did not affect the ratio of respiration to photosynthesis but HTHC reduced it from $27.8 \pm 1.6\%$ to $16.5 \pm 1.3\%$ at the off-shore station  (Fig. 6b). On the contrary, HTHC ($38.7 \pm 3.1\%$) increased the ratio compared to LTLC ($27.3 \pm 2.4\%$), with insignificant effect of temperature alone ($29.5 \pm 3.3\%$) at the near-shore station  (Fig. 6b).

**4 Discussion**

**4.1 Effects of increased temperature and $CO_2$ on biomass**

 The higher temperature increased chl *a* concentration at both stations, which might be attributed to increased active uptake of nutrients at the elevated temperatures through enhanced enzymatic activities. Algal and cyanobacterial growth commonly increases with temperature within a suitable range and then decreases after the optimal temperature point/range (Goldman and Carpenter, 1974; Montagnes and Franklin, 2001; Savage et al., 2004; Boyd et al. 2013) and optimum temperatures for growth of marine phytoplankton are usually several degrees higher than the environmental temperatures (Thomas et al., 2012), which could explained the increase chl a level of phytoplankton grown at the higher temperature

On the other hand, the elevated $CO_2$ offset the positive effect of the higher temperature on chl *a* at the near-shore station , and even reduced chl *a* at the off-shore station . High $CO_2$ can sometimes enhance algal photosynthesis and growth, since $CO_2$ in seawater is suboptimal for full operation of Rubisco enzymes ( Giordano et al.,  2005 and references therein). On the other hand, positive effects of elevated $CO_2$ can be affected by other environmental factors. Gao et al. (2012b) demonstrated that rising $CO_2$ could enhance growth of diatoms at low light intensity, but decrease it at high light intensity. It was found that rising $CO_2$ concentration lowered the threshold for diatom growth above which photosynthetic active radiation becomes excessive or stressful, owing to reduced energy requirements for inorganic carbon acquisition at elevated $CO_2$ (Gao et al., 2012b). In the present study, the mean daily solar radiation levels during incubation were 1312 and 1136 $\mu mol$ photons $m^{-2}$ $s^{-1}$ (Table 1), corresponding to phytoplankton in the microcosms exposed to 1068-1194 $\mu mol$ photons $m^{-2}$ $s^{-1}$ at the off-shore station and 729-1034 $\mu mol$ photons $m^{-2}$ $s^{-1}$ at the near-shore station, which were far above the threshold light intensity reported for diatoms (Gao et al., 2012b). Consequently, the higher $CO_2$ combined with the high solar radiation in summer of NSCS may have imposed negative effects on phytoplankton biomass at both the off-shore and the near-shore station . In addition, the inhibitory effect of higher $CO_2$ on biomass was more significant at the off-shore station  than the near-shore station. This can be attributed to the higher sensitivity of picoplankton to high solar radiation (Li et al., 2011; Wu et al., 2015), which could be delivered to the interaction of high solar radiation and high $CO_2$. As shown in Li et al.'s (2011) study, the proportion of picoplankton in phytoplankton assemblages increased with distance off the coasts. Therefore, the dominant species at the off-shore station  are pico- and nano-phytoplankton, but micro-phytoplankton at the near-shore station  (Table 3).

**4.2 Effects of increased levels of temperature and $CO_2$ on primary productivity**

The seawater volume-specific DPP at the near-shore station  was higher than the off-shore station . This should result from both higher chl *a* concentration and chl *a*-specific DPP at the near-shore station. It has been shown that more smaller cells exist at the off-shore station than at the near-shore station (
[revised manuscript text omitted]
 the two station , negative at the off-shore station and positive at the near-shore station. This can be attributed to differential responses of photosynthesis at both stations to HTHC, considering the responses of respiration were similar.

**5 Conclusions**

This study demonstrates that a short-term rise of SST appeared to simulate the
DPP and dark respiration of phytoplankton assemblages in the NSCS. However, this
positive effect was dampened or offset when warming and ocean acidification
treatments were combined. The regional responses of phytoplankton assemblages at
the two stations to ocean warming and acidification may differ due to differences in
physical and chemical environment as well as phytoplankton community structure.
The combined treatment of warming and acidification reduced biomass and dark
respiration rate at the off-shore, but did not affect them at the near-shore station.
Ecologically and geographically, our data implies differential responses of primary
production to ocean climate change. This short-term experiment suggests the need to
determine whether similar effects may occur over the longer timescales of future
anthropogenic change.

**Acknowledgements**

This study was supported by

[revised manuscript text omitted]

and near-shore station D001. SST: seawater surface temperature; N: $NO_3^- + NO_2$ (μmol

$L^{-1}$); P: $PO_4^{3-}$ (μmol $L^{-1}$). Data of nutrients and phytoplankton composition are derived from literatures.

| Station | SST | Salinity | $pH_T$ | N | P | Dominant phytoplankton |
|---------|-----|----------|--------|---|---|------------------------|
| SEATS | 28.7 | 32.9 | 8.07 | <0.1[a] | <0.01[b] | Pico- and nano-phytoplankton[c] |
| D001 | 26.8 | 33.5 | 8.03 | >1[d] | >0.1[d] | Micro-phytoplankton[e] |

[a]Du et al. (2013); [b]Wu et al. (2003); [c]Li et al. (2011); [d]Li et al. (2014); [e]Zhang et al. (2014).

 **Figure captions**

**Figure 1.** Experimental stations in the northern South China Sea.

**Figure 2.** Chl $a$ concentration in situ (a) and after temperature and $p$CO$_2$ treatments in microcosms (b). The microcosm incubations lasted seven days at off-shore station SEATS and six days at near-shore station D001.The error bars indicate the standard deviations (n = 3). The different letters above the error bars represent significant ($P <$ 0.05) differences between stations in panel (a) and between treatments in panel (b).

**Figure 3.** Daytime primary productivity (DPP) in situ (a, b) and after temperature and $p$CO$_2$ treatments in microcosms (c, d). The microcosm incubations lasted seven days at off-shore station SEATS and six days at near-shore station D001. The error bars indicate the standard deviations (n = 3). The different letters above the error bars represent significant ($P <$ 0.05) differences between stations in panels (a, b) and between treatments in panels (c, d).

**Figure 4.** Net primary productivity (NPP) in situ (a, b) and after temperature and $p$CO$_2$ treatments in microcosms (c, d). The microcosm incubations lasted seven days at off-shore station SEATS and six days at near-shore station D001. The error bars indicate the standard deviations (n = 3). The different letters above the error bars represent significant ($P <$ 0.05) differences between stations in panels (a, b) and between treatments in panels (c, d).

**Figure 5.** Dark respiration in situ (a, b) and after temperature and $p$CO$_2$ treatments in microsoms (c, d). The microcosm incubations lasted seven days at off-shore station SEATS and six days at near-shore station D001. The error bars indicate the standard deviations (n = 3). The different letters above the error bars represent significant ($P <$ 0.05) differences between stations in panels (a, b) and between treatments in panels (c, d).

**Figure 6.** The ratio of respiration to photosynthesis in situ (a, b) and after temperature and $p$CO$_2$ treatments (c, d) in microcosms. The microcosm incubations lasted seven days at off-shore station SEATS and six days at near-shore station D001. The error bars indicate the standard deviations (n = 3). The different letters above the error bars represent significant ($P <$ 0.05) differences between stations in panels (a, b) and between treatments in panels (c, d).

[Figure]

 Figure 1

[Figure]

              Figure 2

[Figure]

Figure 3

[Figure]

Figure 4

[Figure]

Figure 5

[Figure]

Figure 6